# Long Term Follow-Up Observation in Small Choroidal Melanocytic Tumors

**DOI:** 10.3390/cancers16152627

**Published:** 2024-07-23

**Authors:** Laura Prieto-Domínguez, Ciro García-Álvarez, Maria F. Muñoz-Moreno, Patricia Diezhandino, David Miguel-Perez, María A. Saornil

**Affiliations:** 1Ophtalmology Department, University Clinical Hospital of Valladolid, Avenida Ramón y Cajal, 47003 Valladolid, Spain; 2Ophtalmology Department, Ocular Oncology Unit, University Clinical Hospital of Valladolid, Avenida Ramón y Cajal, 47003 Valladolid, Spain; ciro.garcia.alvarez@gmail.com; 3Research Support Unit, University Clinical Hospital of Valladolid, 47003 Valladolid, Spain; mfmunozm@saludcastillayleon.es; 4Radiation Oncology Department, Ocular Oncology Unit, University Clinical Hospital of Valladolid, Avenida Ramón y Cajal, 47003 Valladolid, Spain; pdg22@hotmail.es; 5Radiophysics Department, Ocular Oncology Unit, University Clinical Hospital of Valladolid, Avenida Ramón y Cajal, 47003 Valladolid, Spain; david.miguel@outlook.com

**Keywords:** small choroidal melanoma, observation, survival

## Abstract

**Simple Summary:**

There is ongoing controversy regarding the diagnosis and appropriate treatment of small choroidal melanocytic tumors. Identifying lesions that may evolve into melanomas is crucial due to their metastatic potential. The debate centers on whether it is feasible to delay tumor treatment until signs of progression are evident. This study aims to analyze the long-term outcomes of patients with small choroidal melanocytic tumors under observation, to determine their rate of transformation to melanoma and survival rates in this population. The results suggest that active observation until signs of progression to melanoma appear can be a safe alternative, avoiding the potential side effects of treatment. This approach preserves the visual function and quality of life for patients with tumors that will never progress to melanoma, without compromising their overall prognosis.

**Abstract:**

Background: The purpose of this study is to analyze the long-term evolution of patients with small choroidal melanocytic tumors (SCMTs) undergoing observation, and to assess their rate of transformation into melanomas and survival. Methods: A retrospective single-cohort study of patients with SCMTs (1–3 mm in height and 5–10 mm in base) diagnosed from January 1992 to February 2023 was carried out, with observation as the initial treatment. The main criterion for a transformation into melanoma is considered to be an increase in size of more than 1 mm in height and/or more than 1 mm in base measured on an ultrasound/retinography, recorded in two consecutive visits separated by one to three months. Results: 243 patients were included with a mean age of 65.3 years and a mean follow-up of 7.9 years (6 months–27.9 years); 27 patients showed tumor growth. The probabilities of growth at 5, 10, and 15 years are 10%, 14%, and 17%, respectively. Regarding survival, 22 patients died and only 3 deaths were due to melanoma metastasis. Survival rates at 5 and 10 years are 99% and 97%. Conclusions: Observation is a viable therapeutic option for SCMTs, avoiding the side effects of treatment, considering the majority of these tumors do not progress to melanoma. With close monitoring, patients can be treated promptly upon detecting a transformation. Additionally, the findings confirm that small melanocytic tumors can lead to metastatic disease, albeit at a low rate.

## 1. Introduction

Choroidal melanoma is the most common primary intraocular tumor in adults. It has a relatively low incidence, with approximately 5–9 new cases per million people per year [1,2]. It represents 5% of all melanomas. Despite being a rare condition, it can have serious implications for both the patient’s vision and survival. It is an aggressive tumor with a mortality rate of approximately 50% at 15 years from diagnosis, regardless of the treatment performed [3,4]. 

Tumor size is one of the most important prognostic factors of uveal melanoma [4,5]. Many studies over the last decades have shown that patients with smaller choroidal melanomas at the time of treatment have a better survival prognosis than those with large tumors [6,7,8]. The differences in survival have been attributed to the effectiveness of treatment when it is provided earlier in the natural history of the tumor. The theory proposes that tumor cells enter into the bloodstream, producing metastasis is a time-dependent function; the longer a melanoma is present, the higher the probability of developing metastasis. According to this hypothesis, early treatment (removal or tumor destruction), before intravascular dissemination, can prevent metastasis [9].

There is a controversy regarding the diagnosis and appropriate treatment of small choroidal melanomas. The Collaborative Ocular Melanoma Study (COMS) [10] established the criteria for small melanomas as tumors with a maximum height greater than 1 mm and less than or equal to 2.5 mm, and a maximum base less than or equal to 16 mm. However, without pathological confirmation, some clinically suspected diagnoses of small choroidal melanomas may actually be choroidal nevi. 

There is no clear consensus on establishing the threshold that distinguishes between benign and malignant choroidal melanocytic lesions during the diagnostic process. There are a lack of uniform diagnostic criteria and globally accepted terminology [11]. For this reason, some authors suggest that lesions measuring between 1 mm and 3 mm in height and between 5 mm and 10 mm in base are best referred to as an Indeterminate Choroidal Melanocytic Tumor (ICMT) [12,13,14].

The main concern in the early detection of lesions that evolve into melanomas lies in their potential for metastatic spread regardless of their size [15,16]. This underscores the importance of distinguishing nevi from genuine melanomas. The main indicator of this transformation is the observed growth of the lesion. If an initially diagnosed Indeterminate Choroidal Melanocytic Tumor exhibits signs of growth, the diagnosis of melanoma can be confirmed.

On the other hand, there are risk factors that can be highly valuable in identifying lesions suggestive of malignancy. These factors include size (thickness > 2 mm or diameter > 5 mm), presence of subretinal fluid, decreased visual acuity, presence of orange pigment, and the appearance of kappa angle as an ultrasound finding. The risk of progression to melanoma is directly related to the number of risk factors present in the lesion [5,17].

However, there is no evidence to suggest that the initial treatment of the ICMT reduces its metastatic potential [18]. These treatments can affect visual function, highlighting the importance of avoiding unnecessary treatment of nevi that are unlikely to progress to melanomas. The management of ICMTs remains controversial, as there are limited data on the natural history of these melanocytic lesions [14]. The primary reason for treating these tumors is to avoid the risk of dissemination during observation. However, there is a notable absence of randomized, prospective trials comparing early treatment with delayed treatment pending the confirmation of a diagnosis through documented growth [19]. The safety of the method of observing until growth remains an unanswered question.

Recently, attempts have been made to estimate the impact on survival of delaying treatment for small melanomas [20] using risk prediction programs such as the Liverpool Uveal Melanoma Prognosticator Online (LUMPO) [21]. The results support the theory that delaying treatment until tumor progression is documented is associated with minimal or no increase in death from metastasis. However, there are currently no studies with a substantial number of participants that conduct a comprehensive analysis of long-term follow-up data.

Over the more than 30-year history of the Intraocular Tumors Unit (ITU) of the University Clinical Hospital of Valladolid, nearly a thousand choroidal melanomas have been diagnosed, with approximately one-third managed through observation [22]. The purpose of the present study is to analyze the long-term evolution of patients with small choroidal melanocytic tumors (SCMTs) undergoing observation, with the aim of finding out their rate of transformation into melanomas and survival in this population. The results may contribute to understanding the natural history of these tumors and to finding out if observation can be a safe therapeutic alternative for small choroidal melanocytic tumors, thus preserving the visual function and quality of life of patients without compromising their life prognosis. 

## 2. Materials and Methods

### 2.1. Study Design

This is a retrospective study of a single cohort of consecutive patients diagnosed at the Intraocular Tumors Unit (ITU) of the University Clinical Hospital of Valladolid with SCMTs subjected to observation as the initial therapeutic option. This study has been conducted following the principles of the Helsinki Declaration. Approval was obtained from the hospital’s research committee for institutional data analysis.

### 2.2. Patients

Patients attending the ITU between January 1992 and February 2023 with SCMTs were included in this study, meeting the following inclusion criteria: measurements between 1 and 3 mm in height and between 5 and 10 mm in base, with observation as initial therapeutic option, and, at least, 6 months of follow-up.

### 2.3. Data Collection

Patients were examined by experienced ocular oncologists (MAS/CGA) by complete ocular examination, including ophthalmoscopy/retinography, ultrasound, and data included in a specific prospective protocol designed for melanoma patients since 1992.

A data review for the present study included the following variables at the initial examination: patient age and sex, laterality, reason for consultation, and ultrasound tumor dimensions (largest tumor basal diameter and maximum height). To measure the tumor via ultrasound, axial, longitudinal, and transverse scans are performed. These scans facilitate measurements of the maximum base, a secondary base at a 90° angle to the first, and the tumor height.

Other tumor features, such as the presence of orange pigment and subretinal fluid, were also assessed by ophthalmoscopic examination and supplemented by ancillary studies such as autofluorescence and optical coherence tomography.

Patients were followed-up with a complete ocular examination every 6 months for the first 5 years, and then once a year if no changes were observed, and only one risk factor was present, and the patient remained asymptomatic.

Additionally, a systemic extension study was conducted on all patients, including a liver profile and an abdominal ultrasound at the time of the initial diagnosis.

### 2.4. Criteria for Tumor Transformation and Treatment

The main criterion for the transformation into melanoma is considered to be an increase in lesion size of more than 1 mm in height and/or more than 1 mm in base measured on an ultrasound, recorded in two consecutive visits separated by one to three months. Other signs of malignancy include tumor growth observed in retinography or an increase to more than two risk factors including the appearance or increase in orange pigment, the presence or increase in subretinal fluid, and decreased visual acuity or tumor growth exceeding 2 mm.

If any of these malignant transformation criteria appear, treatment with episcleral brachytherapy or enucleation was proposed, depending on the tumor’s characteristics. Once treated, the follow-up was intensified with a check-up at three months, followed by semi-annual check-ups for five years, and annual check-ups thereafter. Additionally, periodic extension studies were performed with an abdominal ultrasound and blood tests with a liver profile at every medical review.

## 3. Results

### 3.1. Patients

Of 1072 patients diagnosed with uveal melanoma in the ITU in the recruitment period, 243 patients were diagnosed as SCMTs and underwent observation as the primary therapeutic option.

Epidemiological and clinical characteristics are shown in Table 1. At the time of diagnosis, the mean age of the patients was 65.3 years with a standard deviation of 14.1. Regarding gender distribution, the sample comprises 88 males (36.2%) and 155 females (63.8%). The mean follow-up was 7.9 years (with a minimum of 6 months and a maximum of 27.9 years). 

It was observed that 76.0% of the patients were diagnosed during a routine check-up, while the remaining 24.0% consulted medical experts due to visual symptoms.

The lesions included in this study had a mean base measured by an ultrasound of 7.1 (SD 1.5) by 6.6 (SD 1.3), with a maximum height of 1.9 (SD 0.5).

### 3.2. Growth 

During the mean follow-up of 7.9 years (6 months-27.9 years), 27 out of 243 patients showed tumor malignant transformation. The mean survival time without growth was 7.9 years with a 95% CI (6.82–8.39). The mean survival time up to growth is 1.9 years. Figure 1 shows the follow-up time of the patients; there are 243 patients at the start of the follow-up, 146 with 5 years, 73 patients with 10 years, and 12 patients with 20 years. The probabilities of growth at 5, 10, and 15 years are 10%, 14%, and 17%, respectively (Figure 1).

### 3.3. Risk Factors 

The signs of activity presented by each tumor were recorded, finding a base larger than 5 mm in 100% of the included tumors. The most frequently found risk factor was a height greater than 2 mm (42.4%) (Table 1). A statistically significant association was found with the reason for diagnosis (*p* < 0.001) and between the appearance of orange pigment and the growth of the lesion (*p* < 0.05) (Table 2). 

### 3.4. Mortality 

During the follow-up, 22 out of 243 patients died and only 3 deaths were due to melanoma metastasis. The mean overall survival time was 23.9 years with a 95% CI (22.3–25.4). Figure 2 shows the follow-up time of the patients; there are 243 patients at the start of the follow-up; at 5 years the number of patients at risk is 163 and their cumulative probabilities are 96 and 99% for global and specific survival, respectively. With 10 years of follow-up, there would be 86 patients with cumulative survival probabilities of 92 and 97%, respectively. The specific survival rate remained stable until the end of this study (Figure 2).

### 3.5. Analysis of Patients with a Tumor Transformation into Melanoma

Among the 27 tumors that exhibited growth or experienced an increase in risk factors to more than two, one of them was treated with enucleation (3.7%) due to optic nerve infiltration, while the other 26 were treated with brachytherapy (96.3%). On the other hand, 25 of the 27 tumors were treated due to suspected malignancy based on an increase in size on an ultrasound and/or retinography; only two were treated due to an increase in risk factors. No patient showed recurrence after treatment. The characteristics of these 27 tumors are shown in Table 3.

## 4. Discussion

The results of the present study, analyzing the long term evolution of small choroidal melanocytic tumors, support the theory that they can be effectively managed through observation, given that only 17% progressed to melanoma over the course of the follow-up, with 10% appearing in the first 5 years; a fact that justifies a closer follow-up during this period of time. However, it has also been observed that the remaining 7% can appear at any time during the subsequent years. This approach helps avoid aggressive initial treatment and its potential side effects for the majority of patients. Therefore, a close and prolonged follow-up is essential to detect early signs of a transformation into melanoma and to perform prompt treatment. As emphasized by several authors, standardized ultrasound procedures are essential for the diagnosis and evaluation of progression in these lesions [23]. However it is crucial to note that these small tumors have a metastatic potential in 3% of patients. Twenty two patients died during this study and only three died due to melanoma metastasis (cumulative specific survival 97%), despite the early detection of melanoma transformation and prompt treatment.

The challenge of identifying small choroidal melanomas and determining their appropriate therapeutic approach, without compromising survival, has been a focus for decades among expert ocular oncology groups. Studies have been published on these tumors, investigating their growth and survival, but comparing them is difficult due to variability in inclusion criteria, methodologies, and follow-up durations. 

The fact that a small melanoma can be fatal is known, but has only been specifically studied by the European Ophthalmic Oncology group in a multicentric survey [18]. The authors recruited 45 cases from 10 ocular oncology services, with choroidal melanomas < 3 mm height and < 9 mm in the largest basal diameter, who developed metastases in order to assess their characteristics. The authors suggest that 3.0 mm is the size limit for the ability of a choroidal melanoma to metastasize, and that no known clinical characteristics can predict this event.

The Collaborative Ocular Melanoma Study (COMS) is a multicentric set of clinical trials developed in North America in the last decades of the 20th century, which included one arm of patients with small choroidal melanoma (1–3 mm in apical height and 5–16 mm in the largest basal dimension). This is a non-randomized, prospective follow-up study, including 204 patients to describe time to tumor growth and mortality (COMS report No. 4) [10]. The study showed that 21% of cases demonstrated growth by 2 years, and 31% by 5 years. Specific mortality at 1 and 8 years was 1% and 3.7%, respectively. These results show similar data to the present study in terms of mortality, but a higher proportion of tumor growth (31% vs. 10% at 5 years). These differences can be attributed to the different growth criteria applied, but also to the racial and geographic differences of the populations studied, as it is a recognized fact that these factors influence both the incidence and evolution of patients with uveal melanoma [1,2,4,8].

Singh and collaborators have carried out extensive research on terminology or diagnostic criteria for small choroidal melanomas, as well as the impact of deferring treatment until the diagnosis is confirmed [11,12,13,19]. In one of their last papers [12], they studied 167 patients with a small choroidal melanoma (size 5–16 mm in the largest basal diameter and 1–2.5 mm in height) to quantify potential loss of vision and freedom from metastasis after a period of surveillance to documented growth (42 patients) vs. immediate treatment (125 patients) with a median follow-up of 34.6 months. The authors applied a prediction model to obtain the predicted risk of melanoma (high vs. low), and they concluded that a low-risk choroidal melanoma can be managed by surveillance to documented growth before receiving vision-threatening treatment without an increased risk of metastatic death.

Damato et al. developed models for predicting mortality from choroidal melanoma, and they estimated the metastatic death risk when the treatment of small choroidal melanomas is deferred until growth is observed [20,21]. They include 24 patients with an exponential growth rate estimated at 4.3% per year. Using the Liverpool Melanoma Prognosticator Online (LUMPO3), they measured changes in 15-year metastatic and non-metastatic death risks according to deferral until demonstrated growth or immediate treatment; the authors conclude that deferring treatment is associated with minimal increase in metastatic mortality.

Vigué-Jorba et al. [24] analyze the metastatic survival rates in patients with small choroidal melanocytic lesions (<3 mm height and <12 mm base by an ultrasound) initially observed, and who showed progression during the follow-up (increase of >0.3 mm base >0.5 mm in height). They include 75 patients with a mean follow-up of 81 m. At diagnosis, a median of five risk factors was present, and the median time until growth was 17 months postdiagnosis; then, tumors were treated. Melanoma specific survival was 98% at 5 years and 91.6% at 10 years. Comparing this series of patients from the same country as our study, which were more similar in terms of racial and geographical characteristics, they found a higher growth rate in the first years (similar to the present study). Regarding survival at 5 and 10 years, this seems to be higher, but this may be because this series of cases only includes those who grew and were treated for melanomas, that is, only confirmed melanomas.

Wu et al. [25] studied a group of 54 patients with melanomas that were previously observed as indeterminate lesions, with the objective of determining growth rates calculated as change in lesion thickness in mm per month. The authors found that the mean growth rate appears to be relatively slow during the period of observation, and it accelerates at the time of melanoma diagnosis.

Due to the differences in the inclusion criteria of patients in the aforementioned studies, it is difficult to compare the results with the present study. The present study includes patients with a base of 10 mm or less, while the majority include lesions with a base of up to 16 mm [10,12,24]. There are also differences in the follow-up time, with the present study being one of the largest series with the longest follow-up time with 7.9 years on average. These facts, together with the demographic differences between the patient series, may explain the discrepancies in the results.

Regarding risk factors for melanoma development, among the 27 patients who exhibited signs of malignant transformation, the majority (25) demonstrated growth, while only 2 were treated based on increased risk factors. This underscores that ophthalmoscopic and/or ultrasound-detected growth is the primary indicator of a transformation into melanoma.

Clinical risk factors, predictive of growth have been widely studied by the Shields group [5,15,17] in order to provide indirect evidence for small choroidal melanomas. These factors include size, symptoms, orange pigment subretinal fluid and ultrasound characteristics [5,15,17]. While risk factors such as subretinal fluid and orange pigment are considered as a high risk for predicting lesion growth/activity, the demonstration of growth and the observed growth rate is considered evidence of a transformation into melanoma. This fact has been proven by Raval et al. [26], who performed biopsies using fine-needle aspiration biopsies in 30 patients with demonstrated tumor growth prior to treatment, confirming that choroidal melanocytic lesions with confirmed growth can be clinically diagnosed as small choroidal melanomas without the need to perform a biopsy. In our study, we included very small lesions from eyes with good visual function. In these cases, the benefits of fine-needle aspiration biopsy, considering the side effects versus obtaining sufficient diagnostic material, did not justify performing the procedure routinely, given that more than 85% of the lesions did not exhibit growth or malignancy.

Although this research contributes to addressing whether it is safe to observe SMCTs until growth, there remains a lack of randomized, prospective trials comparing prompt versus deferred treatment in these patients.

## 5. Conclusions

The results of the present study indicate that observation is a viable therapeutic approach for small melanocytic choroidal tumors, avoiding the potential side effects of treatment, especially since the majority of these tumors do not progress to melanoma. Close monitoring allows for timely intervention upon detecting signs of a transformation into melanoma, emphasizing the importance of a long-term and consistent follow-up. Furthermore, the findings affirm that small melanocytic tumors can indeed metastasize, albeit at a low rate.

## Figures and Tables

**Figure 1 cancers-16-02627-f001:**
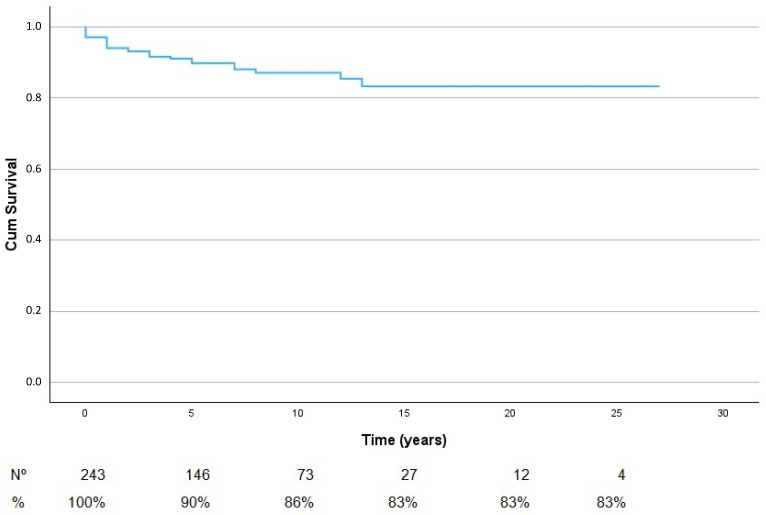
Kaplan–Meier survival curve displaying tumor growth for 243 patients under observational treatment.

**Figure 2 cancers-16-02627-f002:**
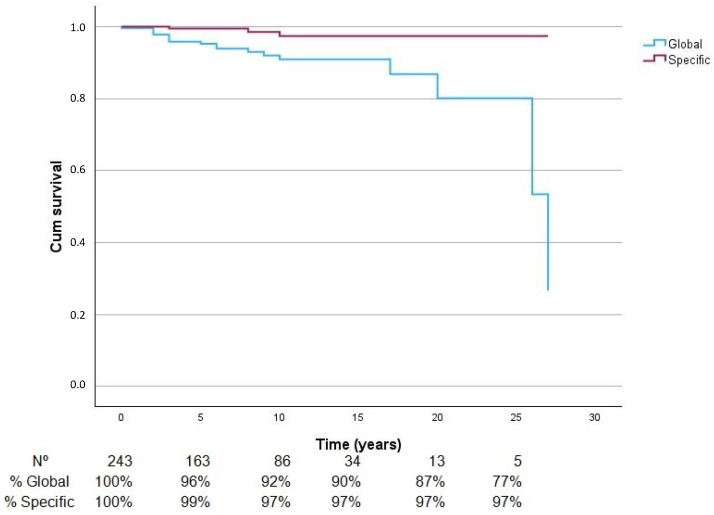
A Kaplan–Meier survival curve showing the overall and melanoma-specific survival of patients under observation.

**Table 1 cancers-16-02627-t001:** Epidemiological profile, clinical characteristics, and signs of activity.

		Mean (SD *)	n	%
**Age**		65.3 (14.1)		
**Gender**	Male		88	36.2%
	Female		155	63.8%
**Follow-up**	Years	7.9 (5.9)		
**Reason for ** **diagnosis**	Routine check-up		184	76.0%
	Visual symptoms		58	24.0%
**Ultrasound**	Maximum base	7.1 (1.5)		
	Base at 90°	6.5 (1.3)		
	Maximum height	1.9 (0.5)		
**Risk factors**	Orange pigment		52	21.6%
	SRF **		41	16.9%
	Decrease in visual acuity		17	9.2%
	Height > 2 mm		103	42.4%
	Base > 5 mm		243	100.0%

* SD: standard deviation and ** SRF: subretinal fluid.

**Table 2 cancers-16-02627-t002:** Association of signs of activity with lesion malignancy.

Correlation between Risk Factors and Malignant Transformation					
	**No**		**Yes**		
	**N = 216**		**N = 27**		
	**N**	**%**	**N**	**%**	** *p* ** **-value**
**Sex**					0.6
Male	77	87.5	11	12.5	
Female	139	89.7	16	10.3	
**Reason for diagnosis**					<0.001
Routine Exam	171	92.9	13	7.1	
Visual symptoms	45	77.6	14	22.4	
**Orange pigment**					0.01
No	173	91.5	16	8.5	
Yes	41	78.8	11	21.2	
**Subretinal Fluid**					0.2
No	181	90	20	10.0	
Yes	34	82.9	7	17.1	
**Height > 2 mm**					0.1
No	129	92.1	11	7.9	
Yes	87	84.5	16	15.5	

The chi-square statistic is significant at the 0.05 level.

**Table 3 cancers-16-02627-t003:** Measurements, risk factors, treatment, and mortality of the 27 cases showing a transformation into melanoma.

Patient	Ultrasound B × H * (mm)	Ultrasound B × H (mm) (6 mo Prior)	Ultrasound Growth	Retinography Growth	>2 Risk Factors	Treatment	Dead Due to Metastases
**1**	Infiltrative	Infiltrative	No	Yes	Yes	Enucleation ^+^	No
**2**	7.8 × 3.8	7.7 × 3.0	No	Yes	No	BT	Yes
**3**	5.1 × 1.9	2.8 × 1.2	Yes	Yes	No	BT	No
**4**	8.6 × 4.6	7.3 × 3.8	Yes	No	Yes	BT	No
**5**	7.6 × 2.3	7.3 × 2.3	No	No	Yes	BT	No
**6**	6.6 × 2.4	6.7 × 1.3	Yes	No	Yes	BT	No
**7**	10.1 × 2.8	8.5 × 2.3	Yes	Yes	No	BT **	Yes
**8**	8.5 × 3.0	5.0 × 3.0	Yes	Yes	No	BT	No
**9**	11.0 × 4.5	9.3 × 3	Yes	No	No	BT	No
**10**	5.3 × 2.1	6.2 × 2.1	No	Yes	Yes	BT	No
**11**	12.4 × 4.8	8.3 × 2.8	Yes	Yes	No	BT	No
**12**	8.4 × 2.0	7.8 × 2.5	No	Yes	Yes	BT	No
**13**	5.7 × 2.3	5.1 × 1.7	No	Yes	Yes	BT	No
**14**	12.9 × 3.4	10.7 × 3.3	Yes	Yes	No	BT	Yes
**15**	9.7 × 3.3	8.5 × 2.6	Yes	No	No	BT	No
**16**	15.3 × 7.7	8.5 × 2.1	Yes	Yes	No	BT	No
**17**	12.5 × 4.0	10.0 × 3.8	Yes	Yes	No	BT	No
**18**	5.0 × 1.0	5.0 × 1.0	No	Yes	No	BT	No
**19**	11.5 × 4.0	7.9 × 2.0	Yes	Yes	No	BT	No
**20**	10.1 × 3.6	9.3 × 3.3	No	No	Yes	BT	No
**21**	6.6 × 2.5	8.0 × 1.9	No	Yes	Yes	BT	No
**22**	11.0 × 3.4	8.8 × 2.3	Yes	Yes	No	BT	No
**23**	9.0 × 1.7	5.0 × 1.0	Yes	Yes	No	BT	No
**24**	6.6 × 2.1	5.0 × 1.0	Yes	Yes	No	BT	No
**25**	8.4 × 2.3	6.7 × 2.1	Yes	No	Yes	BT	No
**26**	7.41 × 2.0	6.2 × 2.1	Yes	No	Yes	BT	No
**27**	15.2 × 9.1	6.4 × 2.0	Yes	Yes	No	BT	No

* B × H: base × height; ** BT: brachytherapy; and ^+^ enucleation due to invasion of the optic nerve.

## Data Availability

The original contributions presented in this study are included in the article; further inquiries can be directed to the corresponding authors.

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
