# Peer review of "Long Term Follow-Up Observation in Small Choroidal Melanocytic Tumors"

_cancers, 2024, doi:10.3390/cancers16152627_

Round 1

Reviewer 1 Report

Comments and Suggestions for Authors

I read this article with great interest and appreciate the authors' effort in writing this interesting article, however, to avoid misunderstandings about the ultrasound technique, the description of the Standardized Ocular Ultrasound is mandatory. To date, many authors have demonstrated the impact of this technique for diagnosis and evaluation of progression and this topic is mandatory before publication. Capasso L, Gioia M, De Bernardo M, Rosa N. Comment on Roelofs, K.A.; et al. Detection of progression of melanocytic choroidal tumors by sequential imaging: is ultrasonography necessary? Tumors 2020, 12, 1856. Tumors (Basel). 2021 March 15;13(6):1306. doi: 10.3390/cancers13061306. PMID: 33804046; PMC ID: PMC8000638.

Author Response

Summary
Thank you very much for taking the time to review this manuscript. Please find the detailed responses below and the corresponding revisions/corrections highlighted/in track changes in the re-submitted file

Point-by-point response to Comments and Suggestions for Authors
Comment 1: 
I read this article with great interest and appreciate the authors' effort in writing this interesting article, however, to avoid misunderstandings about the ultrasound technique, the description of the Standardized Ocular Ultrasound is mandatory. To date, many authors have demonstrated the impact of this technique for diagnosis and evaluation of progression and this topic is mandatory before publication. Capasso L, Gioia M, De Bernardo M, Rosa N. Comment on Roelofs, K.A.; et al. Detection of progression of melanocytic choroidal tumors by sequential imaging: is ultrasonography necessary? Tumors 2020, 12, 1856. Tumors (Basel). 2021 March 15;13(6):1306. doi: 10.3390/cancers13061306. PMID: 33804046; PMC ID: PMC8000638.

Response 1:
Thank you very much for your comment. We agree with your observation, and as a result, we have added a new paragraph in the methodology section explaining the ultrasound procedure. Indeed, performing a standardized ultrasound is key for the diagnosis and follow-up of choroidal melanocytic lesions, as well as for detecting their malignancy.

On page 3, lines 127-130, we have detailed the procedure used when performing ocular ultrasounds on these lesions. We have also included information regarding the necessity of performing ultrasound for the diagnosis and monitoring of these lesions in the discussion on page 8, lines 271-272. Additionally, we have included the review of the article you suggested in your comment in the bibliography.

Reviewer 2 Report

Comments and Suggestions for Authors

In this study the authors report on 243 patients that were followed for a mean of 7.9 years to determine the risk of transformation of small pigmented choroidal lesions into choroidal melanomas. They conclude that the most accurate way to determine malignant transformation of a benign lesion is documented growth over a relatively short time. This finding has been well documented in a number of other studies. The editors of Cancers could decide to add this study to that the list, however no new prognostically helpful information would be added to the current body of literature.  I am a sub-specialist in  ophthalmic ultrasound  with experience over many years of observing thousands of choroidal lesions.  I rely heavily upon standardized diagnostic A-scan to determine both tumor thickness and internal reflectivity of lesions. In my experience it is very rare for a tumor to exhibit an increase in thickness without concurrent reduction in internal reflectivity and occasionally the change in reflectivity indicative of malignant transformation occurs without increased thickness. The authors mention the use of ultrasound to document tumor thickness but do not comment upon the value of internal reflectivity as described by Karl Ossoinig. It would be useful for them to review their A-scans and include the description of internal reflectivity in their study.  Also, the authors don't include in their study the use of fine needle biopsy with cytologic and genetic profiling of choroidal lesions. Some authors strongly feel that the ultimate diagnostic criteria for malignant transformation is the use of this technique which is more useful than growth of a lesion. The authors of this study should expand upon this in their discussion.

In summary this study reaffirms what is standard practice in most ophthalmic oncology centers to equate growth of a small pigmented choroidal lesion with malignant transformation.  The Holy Grail is to identify which small choroidal tumors will transform into melanomas with the potential to metastasize and result in death of the patient. In this study the authors report on 3 patients who died of metastatic melanoma who had no distinguishing characteristics compared to the other 24 patients whose lesions were assumed to have transformed into malignant melanomas based on increased growth.  Identifying and treating such patients would be a noteworthy achievement.

The following are grammatical or spelling errors that need to be corrected. I suggest this manuscript be reviewed by an editor fluent in English.

line 32                                     separated by one to three months

line 55                                     when it is provided            The theory proposes

line 80                                     decreased visual acuity

line 101                                  undergone to observation (it's unclear what the                                                   author is trying to say)

line 104                                  may contribute to understanding

line 112                                  as the initial

line 138                                  separated by one to three months

line 188                                  without growth

line 189                                  mean survival time up to growth is 1.9 years (is                                                   this a correct statement?)

line 207                                  growth of the lesion

line 241                                  Remaining stable the specific survival (it is                                                           unclear what the author is trying to say

line 250                                  experience

line 254                                  were treated

line 269                                  prolonged

line 300                                  and the influence prompt (it is unclear what the                                                  author is trying to say)

line 309                                  developed

line 319                                  during follow up

line 324                                  that were more

Comments on the Quality of English Language

Author Response

Summary
Thank you very much for taking the time to review this manuscript. Please find the detailed responses below and the corresponding revisions/corrections highlighted/in track changes in the re-submitted file.

Point-by-point response to Comments and Suggestions for Authors
Comments 1:
In this study the authors report on 243 patients that were followed for a mean of 7.9 years to determine the risk of transformation of small pigmented choroidal lesions into choroidal melanomas. They conclude that the most accurate way to determine malignant transformation of a benign lesion is documented growth over a relatively short time. This finding has been well documented in a number of other studies. The editors of Cancers could decide to add this study to that the list, however no new prognostically helpful information would be added to the current body of literature.

Response 1:
As you point out, we agree that rapid growth is a well-documented indicator in numerous studies for determining the malignancy of a melanocytic lesion. We believe that our study represents the largest series of patients with the longest follow-up to date. Therefore, it provides a deeper understanding of the natural history of melanoma and supports monitoring as a safe strategy for managing these patients.

Comments 2:
I am a sub-specialist in  ophthalmic ultrasound  with experience over many years of observing thousands of choroidal lesions.  I rely heavily upon standardized diagnostic A-scan to determine both tumor thickness and internal reflectivity of lesions. In my experience it is very rare for a tumor to exhibit an increase in thickness without concurrent reduction in internal reflectivity and occasionally the change in reflectivity indicative of malignant transformation occurs without increased thickness. The authors mention the use of ultrasound to document tumor thickness but do not comment upon the value of internal reflectivity as described by Karl Ossoinig. It would be useful for them to review their A-scans and include the description of internal reflectivity in their study. 

Response 2:
We completely agree with your comment. Thank you for highlighting the importance of ultrasound, particularly A-scan. We would have liked to consider internal reflectivity when studying these lesions; however, given that this is a very old series (with patients from over 30 years ago), It would be very difficult to obtain these data retrospectively, making it impossible to gather this information for all patients in this case. We have added a detailed description of the procedure used when performing ocular ultrasounds on these lesions on page 3, lines 127-130. Additionally, we have included information regarding the necessity of performing ultrasound for the diagnosis and monitoring of these lesions in the discussion on page 8, lines 271-272. Nevertheless, we will take your advice into account and include A-Scan in our next study on this topic.

Comments 3:
Also, the authors don't include in their study the use of fine needle biopsy with cytologic and genetic profiling of choroidal lesions. Some authors strongly feel that the ultimate diagnostic criteria for malignant transformation is the use of this technique which is more useful than growth of a lesion. The authors of this study should expand upon this in their discussion.

Response 3:
Another very interesting comment. A fine needle biopsy for cytology and genetic profiling was not performed primarily because this study includes very old patients treated when these techniques were not yet available. On the other hand, we included very small lesions from eyes with good visual function. In these cases, we believe that the benefits of fine needle aspiration biopsy (considering side effects versus obtaining sufficient diagnostic material) did not justify performing the procedure routinely, given that more than 85% of the lesions do not exhibit growth or malignancy. We have added a paragraph to the discussion to clarify this issue according to your suggestion on page 10, lines 356-360.
Additionally, we have considered authors such as Singh that correlate genetic alterations with tumor growth, concluding that biopsy is unnecessary to diagnose lesions exhibiting growth as melanoma. (Vishal Raval, Shiming Luo, Emily C. Zabor, Arun D. Singh; Small Choroidal Melanoma: Correlation of Growth Rate with Pathology. Ocul Oncol Pathol 17 December 2021; 7 (6): 401–410)

Comments 4:
In summary this study reaffirms what is standard practice in most ophthalmic oncology centers to equate growth of a small pigmented choroidal lesion with malignant transformation.  The Holy Grail is to identify which small choroidal tumors will transform into melanomas with the potential to metastasize and result in death of the patient. In this study the authors report on 3 patients who died of metastatic melanoma who had no distinguishing characteristics compared to the other 24 patients whose lesions were assumed to have transformed into malignant melanomas based on increased growth.  Identifying and treating such patients would be a noteworthy achievement.

Response 4:
Indeed, it would be very interesting to identify the characteristics that differentiate the tumors of the three patients who died from metastasis from the other lesions. As you previously mentioned, a genetic study of these lesions could have been insightful. However, some of these patients passed away a long time ago, making such studies impossible. Future research aimed at identifying the tumor characteristics that contribute to greater aggressiveness would be valuable. We will consider your suggestion for our future work.

Response to Comments on the Quality of English Language
Thank you for your detailed reading of the text. We have made the modifications to the grammatical errors you suggested and will review for other possible mistakes.